# Evaluation of Chemical Parameters of Urban Drinking Water Quality along with Health Risk Assessment: A Case Study of Ardabil Province, Iran

**DOI:** 10.3390/ijerph18105179

**Published:** 2021-05-13

**Authors:** Reza Aghlmand, Saeed Rasi Nezami, Ali Abbasi

**Affiliations:** 1Department of Civil Engineering, Faculty of Engineering, Ferdowsi University of Mashhad, Mashhad 9177948974, Iran; rezaaghlmandcivil@gmail.com; 2Department of Civil Engineering, Faculty of Engineering, University of Mohaghegh Ardabili, Ardabil 5619911367, Iran; rasinezami@uma.ac.ir; 3Water Resources Section, Faculty of Civil Engineering and Geosciences, Delft University of Technology, 2628 CN Delft, The Netherlands

**Keywords:** health risk assessment, water quality index, Parsabad city, carcinogenic/non-carcinogenic risk, trace elements

## Abstract

In recent years, in addition to water resources’ quantity, their quality has also received much attention. In this study, the quality of the urban water distribution network in northwestern Iran was evaluated using the water quality index (WQI) method. Then, some important trace elements were investigated, and finally, the health risk assessment was evaluated for both carcinogenic elements (Ni, Cd, Cr, Pb, and As) and non-carcinogenic elements (Ca, Mg, Na, K, F, NO_3_, and Cu) using carcinogenic risk (CR) and hazard quotient (HQ), respectively. In the present study, the WQI was calculated based on both World Health Organization (WHO) and Iranian drinking water standards. Comparing the results of these standards revealed that the WQI based on the Iranian standard was slightly higher. Regarding the calculated WQI for the study region, the status of water quality for drinking consumption is in the good water quality class (25 < WQI < 50). It was observed that Cu and Cd have the highest and lowest concentrations in all sampling points, respectively. Hazard Index (HI) results showed that the non-carcinogenic substances studied had a low risk for both adults and children (<1.0). However, the CR results showed that Ni, Cd, and As were above the desired level for both children and adults. The results of this study can be applied for efficient water management and human health protection programs in the study area.

## 1. Introduction

Access to clean drinking water is a fundamental human right, regardless of color, religion, nationality, wealth, or belief. Contaminated drinking water as well as poor sanitation are associated with the transmission of diseases such as diarrhea, polio, cholera, and dysentery. Globally, at least two billion people use fecal-contaminated drinking water sources [1,2]. The increasing need for energy, food, and housing as a result of population growth, urbanization, and modernization creates great pressure on water resources, especially water quality, as well as problems of sewage disposal and contamination of surface waters. Water quality, by definition, is a criterion for assessing the usability of water for different purposes (drinking, industry, agriculture, etc.) using various biological, physical, and chemical parameters [3,4].

Parsabad city is considered an agricultural hub in Ardabil province, Iran, that plays a key role in ensuring food security on a national scale, especially in northwestern Iran. Using chemical fertilizers and chemical pesticides in this area has caused serious water quality issues in the study area. As the effects of draining agricultural water on the quality of drinking water in Parsabad have not been assessed so far, the results of the present study would be useful.

The Aras river is the only source of drinking water in Parsabad city. The lack of accurate identification of dangerous pollutants in this water source and subsequent lack of control in the drinking water treatment plant of this city can threaten the health of the inhabitants. Therefore, water quality monitoring through sampling from the water distribution network in critical points of the city is mandatory. These samples can be used to evaluate the water quality.

In this study, the water quality index (WQI) method has been used to evaluate the quality of drinking water in Parsabad city. The goal of this approach was to give a single value to water quality, which is calculated by considering a list of parameters and constituents [5]. The WQI method is widely used in assessing the quality of surface water and groundwater resources and plays a very important role in water resources’ management [6]. In the present study, WQI was calculated in two ways: (a) using the World Health Organization (WHO) drinking water standard, and (b) using the Iranian drinking water standard. Then, the results of both were presented and compared.

Considering the importance of the drinking water resources used by local residents and the prevalence of various gastrointestinal cancers in the region, as one of the possible consequences of contaminated water consumption, it is necessary to identify different pollutants in the drinking water source. These results can lead to an accurate water quality evaluation in Parsabad city and, consequently, achievement of integrated and sustainable management of water resources in this region. In this study, the health risk assessment was evaluated from both approaches of carcinogenic and non-carcinogenic health risk assessment using carcinogenic risk (CR) and hazard quotient (HQ). In assessing health risk, both ingestion and dermal absorption effects were taken into account. Additionally, an attempt was made to localize health risk calculations; in other words, the parameters included in the health risk indices are consistent with the study area. Therefore, instead of using default values (as in previous studies) in the health risk calculations, we tried to use values calibrated based on the conditions and criteria of the study area.

## 2. Materials and Methods

### 2.1. Study Area and Sampling

Parsabad city, a border city in Ardabil province, northwest Iran, is located 230 km north of Ardabil city and along the Aras border river, from 39°12′ N to 39°42′ N latitude and 47°10′ E to 48°21′ E longitude, as shown in Figure 1. Surface water is the most important source of water in this region. The Aras river is the most important water source for Parsabad city. As one of the largest rivers in northern Iran, with a length of approximately 1072 km, the Aras river begins from Turkey and then crosses the common border of Iran with Armenia and Azerbaijan, and finally flows into the Caspian Sea. Parsabad city, with a population of about 177,601 people, is the second largest city in Ardabil province. This region is one of the most important agricultural hubs in Iran, with the production of more than 50 types of crops, which meet about 80% of Iran’s need for corn seeds.

Therefore, due to the existence of large agricultural lands in the entire outskirts of the city and the use of chemical fertilizers in these lands, there is a possibility of contamination of the drinking water resources in this city, and therefore, the quality of the drinking water source in this area should be surveyed.

### 2.2. Water Quality Index Method

Water quality indices have been used in recent years to evaluate the quality of water consumed by humans in various studies [7,8,9,10,11,12]. Water quality indices can be considered water quality models, as a simplified representation of a complex reality [13]. These indices provide a comprehensive picture of both surface water and groundwater quality for different purposes (e.g., domestic use and irrigation) [14]. In this study, a common and basic formula called the water quality index (WQI) was used [15,16,17,18]. To calculate the WQI, various water quality parameters measured in the year 2019, including pH, HCO_3_, Cl, SO_4_, NO_3_, F, Ca, Mg, K, and Na, were used. It should be noted that the World Health Organization [19,20] and Iranian standards [21] for drinking purposes were considered to calculate the WQI. A minimum weight of 1 was assigned to low-significance parameters in the water quality assessment and a maximum weight of 5 was assigned to parameters with high importance (Table 1). In general, the WQI is computed using Equations (1)–(4) as follows:
(1)$$W_{i}=\frac{w_{i}}{\sum_{i=1}^{n}w_{i}}$$
(2)$$q_{i}=\frac{C_{i}}{S_{i}}\times 100$$
(3)$${\mathrm{SI}}_{i}{=W}_{i}{\times q}_{i}$$
(4)$$\mathrm{WQI}=\sum_{i=1}^{n}{\mathrm{SI}}_{i}$$
where W*_i_* is the relative weight, *w_i_* is the weight of each parameter, *n* is the number of parameters, $q_{i}$ is the water quality rating, $C_{i}$ is the measured concentration of each parameter, $S_{i}$ indicates the drinking water standard for each parameter (mg/L), and ${\mathrm{SI}}_{i}$ is the sub-index of the *i*-th parameter.

The classification of water quality based on the WQI value is represented in Table 2. Accordingly, water quality is divided into five general classes, which include excellent quality, good quality, poor quality, very poor quality, and unsuitable for drinking purposes [22,23].

### 2.3. GIS Application

To estimate the value of required parameters in ungauged areas, sampling points should be used through interpolation. In the present study, geographic information system (GIS) software was used for the spatial interpolation of various water quality parameters and for the preparation of distribution maps for each parameter in the study area. The results of comparing different interpolation methods in GIS show that the kriging method, particularly regression kriging, has a much better performance than other methods, such as inverse distance weighting (IDW) and spline [25]. Therefore, in this study, the kriging method in GIS software was used to prepare spatial distribution maps of water quality parameters.

### 2.4. Health Risk Assessment

Trace elements, particularly some heavy metals, are non-degradable, resistant, and often recycled through biological and physicochemical processes that can pose a significant threat to human health by damaging the nervous system and other internal organs [26]. In recent years, many researchers have tried to assess the potential hazard of trace elements in water to human health using existing methods [27]. In general, human exposure to trace elements can occur through three main pathways: direct ingestion, inhalation through the mouth and nose, and dermal absorption. In the water environment, ingestion and dermal absorption are more important and common [28], and the exposure dose from the two mentioned pathways can be calculated using Equations (5) and (6), which are adapted from the risk assessment guidance by the United States Environmental Protection Agency [29,30].
(5)$${\mathrm{ADD}}_{\mathrm{ingestion}}=\frac{C_{w}\times \mathrm{IR}\times \mathrm{EF}\times \mathrm{ED}}{\mathrm{BW}\times \mathrm{AT}}$$
(6)$${\mathrm{ADD}}_{\mathrm{dermal}}=\frac{C_{w}{\times \mathrm{SA}\times K}_{P}{\times \mathrm{ET}\times \mathrm{EF}\times \mathrm{ED}\times 10}^{-3}}{\mathrm{BW}\times \mathrm{AT}}$$
where ADD_ingestion_ and ADD_dermal_ are the average daily exposure doses through ingestion and dermal absorption of water (mg/kg/day or µg/kg/day), respectively; C_w_ is the average concentration of trace elements in water (mg/L or µg/L); IR is the ingestion rate (L/day); EF is the exposure frequency (day/year); ED is the exposure duration (year); SA is the exposed skin area (cm^2^); K_P_ is the dermal permeability coefficient in water (cm/h)—in this study, 0.0001 for Pb [26], 0.002 for Cr, 0.001 for As, Cd, Cu, Ca, Mg, Na, K, F, and NO_3_, and 0.0002 for Ni [29,31]; ET is the exposure time (h/day); BW is the body weight (kg); and AT is the averaging time (day). The default values assigned for the above variables are shown in Table 3 [30,32,33].

In this study, in order to characterize carcinogenic and non-carcinogenic risk, Equations (7)–(9) were used [26]. The potential non-carcinogenic and carcinogenic risks were evaluated using hazard quotient (HQ) and carcinogenic risk (CR), respectively. The Hazard Index (HI) represents the total non-carcinogenic risks of trace elements from all applicable pathways (e.g., ingestion and dermal absorption). If HQ or HI < 1, the non-carcinogenic health risk is low, but if HQ or HI > 1, non-carcinogenic effects should be considered. The acceptable range of CR according to the United States Environmental Protection Agency (USEPA) is 10^−6^ to 10^−4^ [30].
(7)$$\mathrm{Hazard}\;\mathrm{Quotient}\;(\mathrm{HQ})=\frac{\mathrm{ADD}}{\mathrm{RfD}}$$
(8)$$\mathrm{Hazard}\;\mathrm{Index}\;(\mathrm{HI})=\overset{n}{\underset{i=1}{\sum }}\left({\mathrm{HQ}}_{\mathrm{ingestion}}+{\mathrm{HQ}}_{\mathrm{dermal}}\right)$$
(9)Carcinogenic Risk (CR)=ADD×CSF
where ADD is the average daily exposure dose through ingestion or dermal absorption (mg/kg/day or µg/kg/day); RfD is the reference dose (mg/kg/day or µg/kg/day); CSF is the cancer slope factor of a carcinogen/trace element, (µg/kg/day)^−1^ or (mg/kg/day)^−1^. CSF values were extracted from previous studies [34,35,36,37,38,39,40]. In this study, unlike previous studies in this field, the values of RfD_ingestion_ and RfD_dermal_ for the study area have been localized/specialized. The equations used to calculate RfD_ingestion_ and RfD_dermal_ are as follows:
(10)$${\mathrm{RfD}}_{\mathrm{ingestion}}=\frac{C_{\mathrm{IS}}}{W_{m}}{\times P}_{\mathrm{ingestion}}$$
(11)$${\mathrm{RfD}}_{\mathrm{ingestion}}=\frac{C_{\mathrm{IS}}}{W_{m}}{\times P}_{\mathrm{dermal}}$$
where $C_{\mathrm{IS}}$ is the Iranian standard value for each parameter, $W_{m}$ is the mean weight of a person, $P_{\mathrm{ingestion}}$ indicates the total per capita water consumption for cooking and drinking in Iran, $P_{\mathrm{dermal}}$ shows the total per capita water consumption for bathing, showering, washing, and sanitation in Iran. The average $P_{\mathrm{ingestion}}$ in Iran is about 11 L/day and the average $P_{\mathrm{dermal}}$ is about 87.5 L/day [41]. In this study, $W_{m}$ was considered to be 45 kg.

## 3. Results and Discussion

The spatial distribution maps of different water quality parameters are shown in Figure 2. It can be observed that the parameters of K, Mg, Na, HCO_3_, SO_4_, Cr, and Cl are higher mainly in the eastern part of the study area. The values of the other parameters are distributed generally unbalanced all over the desired area, which can be seen in Figure 2.

The WQI was calculated to evaluate the suitability of the urban water quality of the study area for drinking purposes. In this study, the physical and chemical parameters considered in the WQI calculations are pH, calcium, magnesium, sodium, potassium, chloride, sulfate, nitrate, bicarbonate, and fluoride. To calculate the WQI, data from 17 sampling points were used. The WQI was examined based on two standards (i.e., WHO and Iranian standards), the results of which are shown in Figure 3 and Figure 4. The results of the WQI show that by entering the Iranian standard values in the WQI calculations compared to those of the WHO, the WQI in the whole study area is higher. However, there is not much difference between the WQI results when entering the two mentioned standards in the calculations, and according to the classification in Table 2, the WQI value in the whole study region (in both standards) is in the “Good” water quality class (26 < WQI < 50). The spatial distribution maps of the WQI were prepared using the calculation of the WQI at the sampling points of the urban water distribution network followed by interpolation using the kriging method in the GIS environment.

As shown in Figure 3, generally, in the eastern half of the study area, the WQI value is higher than that in the western half, but it is not high enough to change the status of the water quality according to Table 2. It can be observed that the WQI value using the Iranian standard in all sampling points is higher than the WQI value using the WHO standard, according to Figure 4.

The total concentrations of the trace elements in the sampling points ranged from 31.34 to 51.01 μg/L, with a mean value of 39.43 μg/L (Figure 5). It was observed that copper (Cu) has the highest concentration in all sampling points. After Cu, the next highest concentrations belong to Cr, As, Pb, and Ni in the study area. In addition, cadmium (Cd) has the lowest concentration. In the present study, due to the lack of trace elements data in some sampling points, only nine sampling points’ data were used.

In general, 44% of the total concentration of trace elements belonged to Cu, 16% to Cr, 15% to As, 13% to Pb, 11% to Ni, and 2% to Cd. As shown in Figure 5a, sampling points 4 (51.01 μg/L) and 5 (48.69 μg/L) showed the highest total concentrations of trace elements. In this study, in order to assess non-carcinogenic and carcinogenic risk, hazard quotient (HQ) and carcinogenic risk (CR) were used, respectively. The parameters considered in the non-carcinogenic risk calculations were Ca, Mg, Na, K, F, NO_3_, and Cu; the parameters used in the carcinogenic risk assessment included As, Cr, Pb, Ni, and Cd. The results of HQ and HI and those of CR are shown in Table 4 and Table 5, respectively.

The results show that the HI values for all parameters considered in the health risk assessment of non-carcinogenic trace elements are lower than one, and therefore, there is little risk associated with ingestion and dermal absorption in the study area. However, the results of the health risk assessment of carcinogenic trace elements show that nickel, cadmium, and arsenic exceeded the acceptable threshold (1.0 × 10^−4^) for both adults and children. In addition, chromium is at an acceptable level for adults (8.50 × 10^−5^), but not for children (1.28 × 10^−4^), and lead is within the acceptable limits for both children and adults.

In this study, it was observed that the general water quality based on the WQI in the study area was good, while the results of the CR index showed that three out of five trace elements are above the defined threshold. This does not indicate conflict in the results, because the nature and purpose of the two mentioned indices are completely different. The purpose of using the WQI is to achieve a general interpretation of water quality, while the purpose of the CR index is to examine the amounts of cancerous substances in the water and whether the amounts of these substances are suitable for children and adults. This study showed that the use of a simple water quality index (e.g., WQI) cannot be a criterion for water quality planning and management. Rather, for the correct and efficient management of water quality, a wide range of quality parameters should be used. In other words, to judge the water quality of an area accurately, it needs to be evaluated from different perspectives by using various water quality indices. 

## 4. Conclusions

In this study, the water quality of Parsabad city in Ardabil province, Iran, was evaluated using the water quality index method. To calculate the WQI, the WHO and Iranian standards were used and then compared. The WQI results were similar for both standards and showed that the water quality in the whole study area is in the good water quality class. However, the use of the Iranian standard in the calculation of the WQI led to relatively higher values, but not high enough to change the status of the water quality. The results also showed that Cu and Cd have the highest and lowest concentrations at all sampling points among the studied trace elements, respectively. Then, the health risk assessment of carcinogenic and non-carcinogenic parameters was performed using two indices, HQ (or HI) and CR. The HI results showed that the non-carcinogenic substances studied had a low risk for both adults and children (<1.0). However, the CR results showed that Ni, Cd, and As were above the desired level for both children and adults. Cr was only in the safe range for adults, and Pb was in the safe range for both groups (adults and children). It should be noted that only the amount of CR related to ingestion is high, and the risk associated with dermal absorption is low for all elements. Therefore, water managers in the study area should make more efforts in planning and managing water quality in Parsabad city to reduce the health risk of the mentioned elements. The results of the present study showed that in order to understand and make accurate judgments about the water quality in an area, water quality should be considered comprehensively from different perspectives and using various indices.

## Figures and Tables

**Figure 1 ijerph-18-05179-f001:**
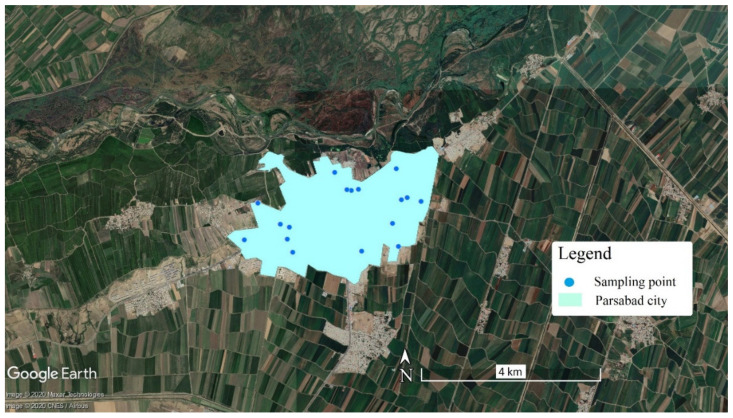
Location of urban drinking water sampling points in the study area.

**Figure 2 ijerph-18-05179-f002:**
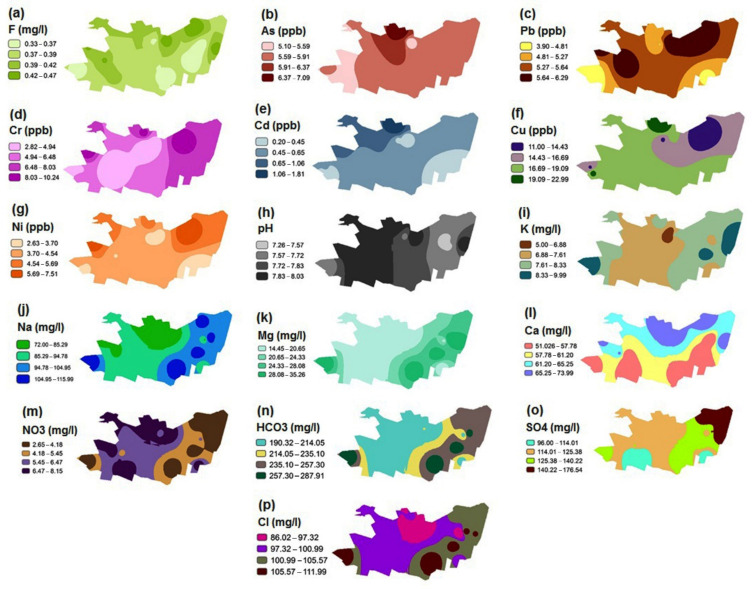
Spatial distribution maps of the various hydrochemical parameters in the study area. (**a**) Fluoride; (**b**) arsenic; (**c**) lead; (**d**) chromium; (**e**) cadmium; (**f**) copper; (**g**) nickel; (**h**) pH; (**i**) potassium; (**j**) sodium; (**k**) magnesium; (**l**) calcium; (**m**) nitrate; (**n**) bicarbonate; (**o**) sulphate; (**p**) chloride.

**Figure 3 ijerph-18-05179-f003:**
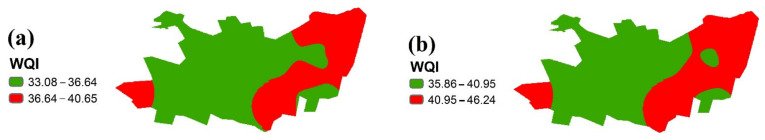
WQI values in the study area using (**a**) the WHO standard and (**b**) the Iranian standard.

**Figure 4 ijerph-18-05179-f004:**
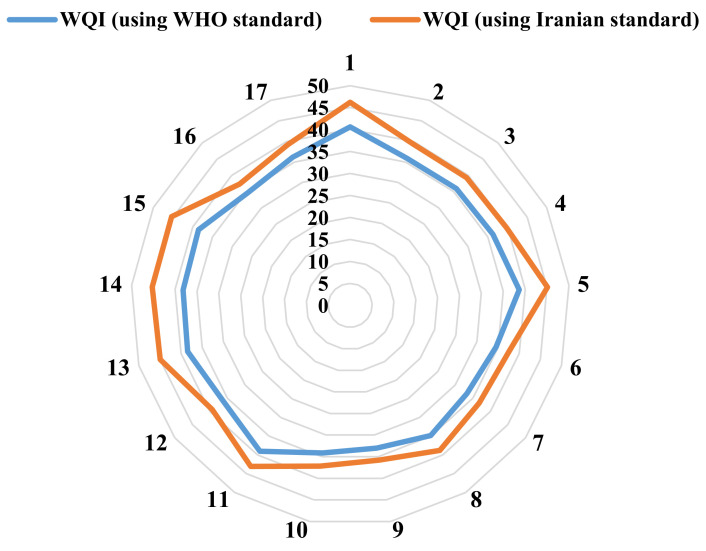
Water quality index (WQI) values in the sampling points.

**Figure 5 ijerph-18-05179-f005:**
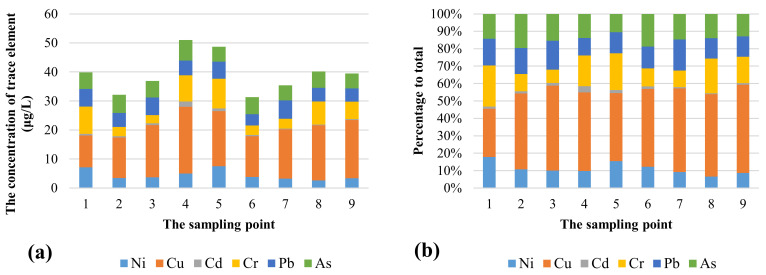
(**a**) Total concentrations of the trace elements at the sampling points (μg/L); (**b**) percentage of various trace elements out of the total concentrations at each sampling point.

**Table 1 ijerph-18-05179-t001:** The weight (*w_i_*), relative weight (W*_i_*), and the standard value of each parameter for both Iranian and WHO standards applied in WQI calculation.

Parameter	Unit	WHO Standard	Iranian Standard	Weight (*w_i_*)	Relative Weight (W_i_)
pH	-	6.5–8.5	6.5–8.5	4	0.1176
HCO_3_	mg/L	300	-	3	0.0882
Cl	mg/L	200–600	250–400	2	0.0588
SO_4_	mg/L	400	250–400	4	0.1176
NO_3_	mg/L	50	50	5	0.1471
F	mg/L	1.5	-	5	0.1471
Ca	mg/L	75–200	300	3	0.0882
Mg	mg/L	50–150	30	3	0.0882
Na	mg/L	200	200	3	0.0882
K	mg/L	10	-	2	0.0588
				∑ *w_i_* = 34	∑ W*_i_*= 1

**Table 2 ijerph-18-05179-t002:** Water quality classification based on WQI values [23,24].

WQI Value	Water Quality Status	Possible Usage
0–25	Excellent	Drinking, irrigation, and industrial
26–50	Good	Drinking, irrigation, and industrial
51–75	Poor	Irrigation and industrial
76–100	Very poor	Irrigation
Above 100	Unsuitable	Proper treatment required

**Table 3 ijerph-18-05179-t003:** Default values in the calculation of average daily exposure dose (ADD) through ingestion and dermal absorption.

Variable	Adults	Children
IR (L/day)	2	0.64
EF (day/year)	350	350
ED (year)	30	6
BW (kg)	70	15
AT (day)	10,950	2190
SA (cm^2^)	18,000	6600
ET (h/day)	0.58	1

**Table 4 ijerph-18-05179-t004:** Reference dose (RfD) (mg/kg/day), average daily exposure dose (ADD) (mg/kg/day), hazard quotient (HQ), and hazard index (HI) for health risk assessment of non-carcinogenic trace elements (subscript “ing”: ingestion; subscript “der”: dermal).

	RfD_ing_	RfD_der_	Adult					Child				
			ADD_ing_	ADD_der_	HQ_ing_	HQ_der_	HI	ADD_ing_	ADD_der_	HQ_ing_	HQ_der_	HI
Ca	73.33	583.33	1.67	0.0088	0.0229	1.5 × 10^−5^	0.0229	2.5077	0.0259	0.0342	4.43 × 10^−5^	0.0342
Mg	7.33	58.33	0.61	0.0032	0.0838	5.5 × 10^−5^	0.0839	0.9181	0.0095	0.1252	1.62 × 10^−4^	0.1254
Na	48.88	388.88	2.56	0.0134	0.0524	3.43 × 10^−5^	0.0524	3.8242	0.0394	0.0782	1.01 × 10^−4^	0.0783
K	2.44	19.44	0.21	0.0011	0.0870	5.71 × 10^−5^	0.0871	0.3177	0.0033	0.1300	1.68 × 10^−4^	0.1301
NO_3_	12.22	97.22	0.14	0.00078	0.0122	8.02 × 10^−6^	0.0122	0.3048	0.0023	0.0249	2.37 × 10^−5^	0.0250
Cu	0.24	1.94	0.00047	2.46 × 10^−6^	0.0019	1.27 × 10^−6^	0.0019	0.00070	7.26 × 10^−6^	0.0029	3.74 × 10^−6^	0.0029
F	0.36	2.91	0.0109	5.67 × 10^−5^	0.0297	1.95 × 10^−5^	0.0297	0.0162	0.00016	0.0443	5.74 × 10^−5^	0.0444

**Table 5 ijerph-18-05179-t005:** Cancer slope factor (CSF) (mg/kg/day)^−1^, average daily exposure dose (ADD) (mg/kg/day), and carcinogenic risk (CR) for health risk assessment of carcinogenic trace elements (subscript “ing”: ingestion; subscript “der”: dermal).

	CSF	Adult					Child				
		ADD_ing_	ADD_der_	CR_ing_	CR_der_	CR_total_	ADD_ing_	ADD_der_	CR_ing_	CR_der_	CR_total_
Ni	0.84	1.21 × 10^−4^	1.26 × 10^−7^	1.02 × 10^−4^	1.06 × 10^−7^	1.02 × 10^−4^	1.81 × 10^−4^	3.73 × 10^−7^	1.52 × 10^−4^	3.13 × 10^−7^	1.52 × 10^−4^
Cd	6.3	1.66 × 10^−5^	8.69 × 10^−8^	1.05 × 10^−4^	5.47 × 10^−7^	1.05 × 10^−4^	2.48 × 10^−5^	2.56 × 10^−7^	1.57 × 10^−4^	1.61 × 10^−6^	1.58 × 10^−4^
Cr	0.5	1.68 × 10^−4^	1.75 × 10^−6^	8.42 × 10^−5^	8.78 × 10^−7^	8.50 × 10^−5^	2.51 × 10^−4^	5.18 × 10^−6^	1.26 × 10^−4^	2.59 × 10^−6^	1.28 × 10^−4^
Pb	0.0085	1.44 × 10^−4^	7.54 × 10^−8^	1.23 × 10^−6^	6.41×10^−10^	1.23 × 10^−6^	2.15 × 10^−4^	2.22 × 10^−7^	1.84 × 10^−6^	1.89 × 10^−9^	1.84 × 10^−6^
As	1.5	1.57 × 10^−4^	8.22 × 10^−7^	2.36 × 10^−4^	1.23 × 10^−6^	2.37 × 10^−4^	2.35 × 10^−4^	2.42 × 10^−6^	3.53 × 10^−4^	3.63 × 10^−6^	3.56 × 10^−4^

## Data Availability

Data available on request due to privacy and ethical restrictions.

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
