# Peer review of "Evaluation of Chemical Parameters of Urban Drinking Water Quality along with Health Risk Assessment: A Case Study of Ardabil Province, Iran"

_ijerph, 2021, doi:10.3390/ijerph18105179_

Round 1

Reviewer 1 Report

General comments:

I enjoyed reviewing the manuscript of “Evaluation of Urban Drinking Water Quality Along with Health Risk Assessment: A Case Study of Ardabil Province, Iran”.

The authors started with the importance of water quality evaluation in Aras River, Iran, and shortly introduced two different evaluation standards used in this study. In general, this paper provided useful information about the water quality in the Aras river area, with both carcinogenic risk and hazard quotient. This study can be applied for efficient water management and health protection.

I recommend the manuscript for publication after minor revisions. Below are some detailed comments and suggestions.

  1. Table 1

For the pH used in the water quality rating calculation, which value of pH used? Is it the average (that is, pH 7.5) or all pH from 6.5 to 8.5 would be account for as qi = 1?

  1. Figure 2

For the figure legend, please use the same size of the letter, some elements like pH (h) and magnesium (k) are too small.

  1. Figure 3

Could you please link the sample point to the map? Or to their spatial distribution? In figure 1, the detailed sample point and their number are not matched, as the result, figure 3 did not give much information on the spatial distribution or the overall behavior about the water quality.

If you just want to show the general WQI, it might be better to list it as a table or to use a bar plot.

Reviewer 2 Report

Dear authors,

Thank you for your interesting paper. It has been written very clearly. I have some small remarks

Line 90: There are large agricultural lands in the entire outskirts, where fertilizers are being used. However, I think also plant protection products or pesticides will be used. Furthermore, it is a long river which will probably pass several cities. As a result, the water will not only contain trace metals, but also organic compounds, which may be a larger problem for drinking water production.

Line 97; … in other words, these indices….

Line 118: WQ1 is used to determine the water quality, e.g. WQ1 = 0-25 then the water quality is excellent. However, only a limited number of compounds (in this case only some trace metals and some general ions like Na and sulfate) take part in this parameter. In other words: excellent water can contain dangerous concentrations of toxic compounds, the standard doesn’t include these. I think this should be mentioned somewhere.

Line 171: is Wm indeed as low as 45 kg?

Reviewer 3 Report

The paper aims to characterise drinking water using the water quality index method. As one of the cited papers (Kachroud et al, 2019) rightly presents, the WQI is liable to misinterpretation if the underlying parameters or the averaging method is not selected carefully. In the present case, beneficial parameters, such as Ca and Mg are evaluated together with parameters posing a risk to health like fluoride. It is missing microbiological quality completely, which is the highest concern in drinking water. There is no information on the drinking water supply of the sampled area, making it difficult to interpret the results. Though the exact concentrations of trace elements are not given, based on Figure 5 it looks like all of them are well below the limit values of the European Union regulation or EPA guide values. Still the calculated carcinogenic risk is extreme (10-4) for three of the 5 heavy metals, clearly suggesting a flaw in the calculation. Carcinogenic risk is calculated for Cr, while there is no indication that Cr(VI) was measured. These errors question the scientific soundness of the study. Data visualisation is nice, but the basis is flawed. 

Round 2

Reviewer 3 Report

.My concerns were not addressed in essence either in the amended text or in the cover letter. The following points still need to be addressed (numbering per the cover letter): 

2. The authors claim that the drinking water supply is described in detail. Important information is missing or can only be inferred from the cover letter: all sampling points are supplied by the same water supply, what is the water treatment technology, is it a piped supply, what kind of taps were sampled etc. Since many of the trace elements are potentially derived from materials in contact with water (Pb, Cu, Ni...) these are important questions to understand variability. 

3.-4. Use of WQI. The WQI used in the study is relevant for surface water quality but not for human health. The basis of evaluation is the relative concentration compared to the limit value. However, some parameters included in the WQI are not contaminants, but essential elements, thus very low concentration is at least as problematic as very high concentration. And I do not see a reason not to include the trace elements of actual health concern into the index. It is not a sufficient justification that similar papers were accepted elsewhere. Most citations use WQI for surface water, not drinking water.

5. No information is given whether Cr(III) or Cr(VI) or both were measured. Cr(III) is a non-carcinogenic essential element. Cr(VI) is a human carcinogen. If the two are not differentiated, the risk calculation is faulty.

6. Acceptable carcinogenic risk level in EU and for USEPA is 10E-06 for non-voluntary risks. Limit values for drinking water are calculated accordingly.

For example:

For Ni, EU limit value is 20 ppb, max value in this study is 7.5 ppb

For Cr, EU limit value is 25 ppb, max value in this study is 10.2 ppb

For Cd, EU limit value is 5 ppb, max value in this study is 1.8 ppb

Yet all amount to (close to) 10E+04 carcinogenic risk. Even if Iranian consumption is 5 times higher than the generally used value (11 L vs 2 L), this is highly unlikely.

SF for these parameters comes from the paper Mohammadi et al, 2019, where further source is not given.
